# Laboratory Cross-Comparison and Ring Test Trial for Tumor *BRCA* Testing in a Multicenter Epithelial Ovarian Cancer Series: The BORNEO GEICO 60-0 Study

**DOI:** 10.3390/jpm12111842

**Published:** 2022-11-04

**Authors:** Zaida Garcia-Casado, Ana Oaknin, Marta Mendiola, Gorka Alkorta-Aranburu, Jose Ramon Antunez-Lopez, Gema Moreno-Bueno, Jose Palacios, Alfonso Yubero, Raul Marquez, Alejandro Gallego, Ana Beatriz Sanchez-Heras, Jose Antonio Lopez-Guerrero, Cristina Perez-Segura, Pilar Barretina-Ginesta, Jesus Alarcon, Lydia Gaba, Antonia Marquez, Judit Matito, Juan Cueva, Isabel Palacio, Maria Iglesias, Angels Arcusa, Luisa Sanchez-Lorenzo, Eva Guerra-Alia, Ignacio Romero, Ana Vivancos

**Affiliations:** 1Molecular Biology Department, Fundacion Instituto Valenciano de Oncologia, 46009 Valencia, Spain; 2Medical Oncology Department, Vall d’Hebron Instituto de Oncologia, 08035 Barcelona, Spain; 3Instituto de Investigacion Biomedica del Hospital La Paz (IdiPAZ), 28029 Madrid, Spain; 4Centro de Investigacion Biomedica en Red de Cáncer (CIBERONC) Instituto de Salud Carlos III, 28029 Madrid, Spain; 5CIMA LAB Diagnostics/Universidad de Navarra, 31008 Pamplona, Spain; 6Molecular Biology Department, Hospital Clinico Universitario Santiago, 15706 Santiago, Spain; 7Fundacion MD Anderson, 28033 Madrid, Spain; 8Departamento de Bioquímica, Instituto de Investigaciones Biomedicas ‘Alberto Sols. Conexion Cancer (UAM-CSIC), Universidad Autonoma de Madrid (UAM), IdiPAZ, 28029 Madrid, Spain; 9Pathology Department, Hospital Universitario Ramon y Cajal, 28034 Madrid, Spain; 10Faculty of Medicine, Alcala University, 28801 Madrid, Spain; 11Instituto Ramon y Cajal for Health Research (IRYCIS), 28034 Madrid, Spain; 12Medical Oncology Department, Hospital Clinico Universitario Lozano Blesa, 50009 Zaragoza, Spain; 13Medical Oncology Department, Hospital Universitario La Paz, 28029 Madrid, Spain; 14Medical Oncology Department, Hospital General Universitario de Elche, 03203 Elche, Spain; 15Universidad Catolica de Valencia, 46001 Valencia, Spain; 16Unidad Mixta de Investigacion en Cancer IVO-CIPF, 46009 Valencia, Spain; 17Medical Oncology Department, Hospital de Sant Pau i Santa Tecla, 43003 Tarragona, Spain; 18Medical Oncology Department, Institut Catala d’Oncologia Girona, 17007 Girona, Spain; 19Medical Oncology Department, Hospital Universitario Son Espases, 07120 Palma de Mallorca, Spain; 20Medical Oncology Department, Hospital Clinic de Barcelona, 08036 Barcelona, Spain; 21Medical Oncology Intercenter Unit, Regional and Virgen de la Victoria University Hospitals, IBIMA, 29010 Malaga, Spain; 22Cancer Genomics Lab, Vall d’Hebron Institute of Oncology (VHIO), 08035 Barcelona, Spain; 23Medical Oncology Department, Hospital Clinico Universitario Santiago, 15706 Santiago, Spain; 24Medical Oncology Department, Hospital Universitario Central de Asturias, 33011 Oviedo, Spain; 25Medical Oncology Department, Hospital Universitario Son LLatzer, 07198 Palma de Mallorca, Spain; 26Medical Oncology Department, Hospital de Terrassa, 08227 Terrassa, Spain; 27Medical Oncology Department, Clinica Universidad de Navarra, 31008 Pamplona, Spain; 28Medical Oncology Department, Hospital Universitario Ramon y Cajal, 28034 Madrid, Spain; 29Medical Oncology Department, Instituto Valenciano de Oncologia, 46009 Valencia, Spain

**Keywords:** ovarian cancer, *BRCA* mutations, NGS, *BRCA* testing, Ring Test Trial

## Abstract

Germline and tumor *BRCA* testing constitutes a valuable tool for clinical decision-making in the management of epithelial ovarian cancer (EOC) patients. Tissue testing is able to identify both germline (g) and somatic (s) *BRCA* variants, but tissue preservation methods and the widespread implementation of NGS represent pre-analytical and analytical challenges that need to be managed. This study was carried out on a multicenter prospective GEICO cohort of EOC patients with known *gBRCA* status in order to determine the inter-laboratory reproducibility of tissue *sBRCA* testing. The study consisted of two independent experimental approaches, a bilateral comparison between two reference laboratories (RLs) testing 82 formalin-paraffin-embedded (FFPE) EOC samples each, and a Ring Test Trial (RTT) with five participating clinical laboratories (CLs) evaluating the performance of tissue *BRCA* testing in a total of nine samples. Importantly, labs employed their own locally adopted next-generation sequencing (NGS) analytical approach. *BRCA* mutation frequency in the RL sub-study cohort was 23.17%: 12 (63.1%) germline and 6 (31.6%) somatic. Concordance between the two RLs with respect to *BRCA* status was 84.2% (g*BRCA* 100%). The RTT study distributed a total of nine samples (three commercial synthetic human FFPE references, three FFPE, and three OC DNA) among five CLs. The median concordance detection rate among them was 64.7% (range: 35.3–70.6%). Analytical discrepancies were mainly due to the minimum variant allele frequency thresholds, bioinformatic pipeline filters, and downstream variant interpretation, some of them with consequences of clinical relevance. Our study demonstrates a wide range of concordance in the identification and interpretation of *BRCA* sequencing data, highlighting the relevance of establishing standard criteria for detecting, interpreting, and reporting *BRCA* variants.

## 1. Introduction

According to Globocan’s 2020 projections, by 2040, the number of women around the world diagnosed with ovarian cancer will rise by almost 37% up to 428,966. The number of women dying from ovarian cancer each year is projected to increase up to 313,617, which is a 50% increase from 2020 to 2022 [1]. Fortunately, the treatment of ovarian cancer has improved significantly in recent years. Today, the use of poly (ADP-Ribose) polymerase inhibitors (PARPis) has been shown to be particularly efficient in epithelial ovarian cancer (EOC) patients harboring germline or somatic *BRCA* mutations (g*BRCA* and s*BRCA*, respectively) [2,3], in addition to tumors with high genomic instability caused by defects in key genes within the homologous recombination (HR) DNA repair machinery [4].

In addition to other genes, both *BRCA1* and *BRCA2* genes play a key role within the HR DNA repair system [5]. Indeed, germline and somatic deleterious mutations in HR genes are identified in approximately 30% of EOC patients, but up to 75% of them are *BRCA1* and *BRCA2* gene mutations [5,6]. Furthermore, *sBRCA* mutations have been described in up to 7% of EOC patients in the first-line or platinum-sensitive relapsed clinical setting [5,7,8].

Consequently, *BRCA* testing in EOC plays a key role in the clinical decision-making process, not only for the identification of familial cancer predisposition but also to personalize therapeutic treatment [9]. The germline testing of *BRCA* is widespread in medical genetics laboratories, but not the identification of patients with *sBRCA* tumors that can benefit from PARPi therapy. Tumor *BRCA* testing on formalin-fixed paraffin-embedded (FFPE) tissue is key to identifying patients with *sBRCA* tumors, which in addition has the advantage of the simultaneous assessment of both somatic and germline mutations using an easily accessible material that is routinely available in any pathology laboratory worldwide. Currently, there is no consensus, but it is recommended to perform both germline and tumor *BRCA* characterization in EOC patients [10,11]. Indeed, *sBRCA* testing is now required to support treatment decisions in many countries, so it is essential that testing is robust [12].

Therefore, high-throughput next-generation sequencing (NGS)-based FFPE-derived DNA sequencing approaches are being implemented, given that they permit fast multiplex testing on small quantities of DNA, improving both the capacity and the cost-effectiveness of mutational analysis compared with traditional methods such as Sanger sequencing [13].

However, not all NGS-based sequencing technology is equal. Both the sequencing error and capacity of the sequencer are important to be considered, as well as data analysis software and the experience that the testing laboratory or center has. For this reason, the Spanish Group of Research on Ovarian Cancer (GEICO) conducted a study to evaluate a range of tumor *BRCA* testing approaches. First, we evaluated the inter-laboratory reproducibility of two Spanish reference laboratories identifying both *gBRCA* and *sBRCA* mutations in 100 FFPE EOC samples. Additionally, second, we determined the efficacy of a spectrum of tumor *BRCA* testing workflows within five Spanish clinical laboratories (CL) to accurately identify *sBRCA* variants in clinical practice in the context of a Ring Test Trial (RTT).

## 2. Materials and Methods

### 2.1. Study Design

The study consisted of a prospective, observational, multicenter study developed at the national level with 30 reference sites in Spain, during a data collection period of 60 months (including follow-up).

The study included 400 patients that met the following inclusion criteria:Written informed consent signed by all patients participating in the study.Histological diagnosis of non-mucinous epithelial ovarian, primary peritoneal, or tubal carcinoma confirmed not more than 2 years prior to the date of ICF signature.Adult women (18 years old or older at the time of diagnosis).Patients may have had other malignancies and have received or are receiving any anticancer therapy, including investigational drugs.Availability of FFPE tumor blocks from the primary tumor for genetic analysis and willingness (100 valid cases).Patients with *gBRCA* testing performed at the site according to current clinical guidelines or willing to be tested centrally if local testing is not available.

Exclusion criteria:Patients without an available medical record (lost, empty, or unretrievable clinical information).

This study has been conducted in accordance with the ethical principles that have their origin in the Declaration of Helsinki (18th World Medical Association General Assembly, Helsinki, Finland) and according to the Spanish Order SAS/3470/2009, and Law 14/2007, of July 3rd, on Biomedical Research. Ethical committee approval (expedient number 025-19), informed consent, cancer family history, and clinical features were collected.

Tumor *BRCA* testing was performed in 100 EOC samples in two reference laboratories (RL1; RL2) to evaluate inter-laboratory reproducibility. The median age at diagnosis of the patients included in our series was 56 years (range 25–84 years). The main characteristics of the patients are included in Table 1. Approximately 16% were g*BRCA* mutation carriers.

### 2.2. Tumor BRCA1/2 Testing Cross-Laboratory Validation

The first 100 valid FFPE tumor blocks from primary tumors were selected to be tested for tumor *BRCA* mutations at RL1. Of these, and after an initial quality control of the samples, 82 valid cases were sent to RL2 for further analysis of tumor mutations of *BRCA*. The workflow of the sample selection strategy is described in Figure 1.

#### 2.2.1. RL1 DNA Extraction

Hematoxylin and eosin (H&E) staining was performed to evaluate tumor cell percentage to guide the macrodissection of the solid tumor sample if ≤30%. A minimum of 10% tumor cellularity was required to be considered. For each sample, DNA was extracted from three unstained sections of 10µm thickness or three 0.6 mm needle biopsies using the QIAamp^®^ DNA Investigator kit (QIAGEN, Hilden, Germany). DNA concentration was quantified using the Quant-iT™ PicoGreen™ dsDNA (ThermoFisher; Waltham, MA, USA) fluorimetric assay.

#### 2.2.2. RL1 NGS

Sequencing was carried out with the Homologous Recombination Solution (HRS; Sophia Genetics) capture kit in the Illumina^®^ MiSeq^®^ sequencer (Illumina; San Diego, CA, USA). The study includes the analysis of the entire coding region and adjacent intronic regions (±25 pb) of 16 genes involved in homologous recombination repair. Bioinformatic analysis of the *BRCA1/2* sequences was undertaken with the analysis software and algorithms developed by Sophia Genetics (Sophia DDM), with the support of other bioinformatic tools such as the Integrative Genome Viewer (IGV; www.broadinstitute.org (accessed on 20 September 2022). The reference genome used for the analysis was hg19. This analysis did not include large rearrangements. The sensitivity limit was set at 5% MAF for point variants and 10% MAF for insertion or deletion variants (indels). The minimum coverage to consider a region properly covered was 600 readings (600x). Pathogenic and likely pathogenic variants were visualized using the IGV software and classification was revised as described in Section 2.4.

#### 2.2.3. RL2 DNA Extraction

Neoplastic content was assessed by H&E staining and a minimum of 20% tumor cellularity was required. For each sample, 5 × 10 µm FFPE sections were cut and genomic DNA (gDNA) was extracted using Maxwell^®^ 16 FFPE Tissue LEV DNA Purification Kit (Promega). DNA was eluted in 60 µL of water and stored frozen at −20 °C.

The sample quality and DNA concentration were determined by fluorimetric quantification using Qubit Fluorimeter and Qubit dsDNA BR Assay Kit (Life Technologies, Carlsbad, CA, USA).

#### 2.2.4. RL2 NGS

DNA libraries were generated from 500 ng of DNA per sample using the SureSelect XT Target Enrichment kit (Agilent Technologies Santa Clara, CA, USA). Briefly, gDNA was fragmented using a Covaris S2 sonicator (Covaris, Woburn, MA, USA) and QC was performed using an Agilent BioAnalyzer 2100 (Agilent Technologies) to ensure an average fragment size of 150 to 200 bp. Fragmentation was followed by end-repair, A-tailing, and sequencing adapter ligation. After 10 cycles of PCR amplification, PCR products were isolated on AMPure XP beads (Agencourt, Beckman Coulter; Pasadena, CA, USA). The prepared library was checked and quantified using an Agilent BioAnalyzer DNA1000 chip and 750 ng was hybridized to a biotinylated probe panel (VHIO-300) containing 430 cancer-related genes. The resulting library was sequenced using the Illumina sequencing by synthesis (SBS) technology (2 × 100 PE run). Sequencing reads were aligned (BWA v0.7.17, Samtools v1.9) against the hg19 reference genome, base recalibrated, indel realigned (GATK v3.7.0, abra2 v2.23) and variant called (VarScan2 v2.4.3, Mutect2 v4.1.0.0). Variants from both callers are reported. A minimum of 5 reads supporting the variant allele was required to identify a mutation. The sensitivity of the technique is 5% MAF for SNVs and 10% MAF for INDELs. Frequent single-nucleotide polymorphisms (SNPs) in the population were filtered based on the gnomAD database (allele frequency 0.0001) and copy number alterations (CNA) were calculated (CNVkit v0.9.6.dev0) but not included in the study. Variants were manually checked and the classification of identified variants was performed using publicly available databases as described in Section 2.4. A minimum 250x on-target coverage was required as part of sample QC.

### 2.3. Variant Analysis

Variant classification and interpretation were carried out using specific databases at the date of issuance of the report, including ClinVar, Breast Cancer Information Core NIH, BRCA Exchange, Leiden Open Variation Database (LOVD), Universal Mutation Database, Human Gene Mutation Database at the Institute of Medical Genetics in Cardiff, COSMIC, cBioPortal, Oncokb and National Library of Medicine and National Institutes of Health (dbSNP). Variants were classified into five categories following the recommendations of the American College of Medical Genetics (ACMG) [14]. The nomenclature used to describe the genetic alterations was that proposed by the Society for Human Genome Variations (HGVS Variant Description Nomenclature) [15]. Reference sequences for *BRCA1* and *BRCA2* were NM007294.3 and NM000059.3, respectively.

### 2.4. Ring Test Trial (RTT)

Out of the 82 cases analyzed by the reference laboratories, 6 were selected by the RLs (based on the type(s) of mutation(s) detected) to conduct the RTT among five laboratories from GEICO hospitals. All laboratories received a total of 9 samples for the tumor testing of *BRCA* mutations, including commercial synthetic (CC_1-3) human FFPE from Horizon (Horizon Dx; Cambridge, UK) (*n* = 3), OC tumor tissue derived DNA (*n* = 3) (tDNA_1-3) and FFPE (*n* = 3) samples (FFPE_1-3) (Figure 1). Laboratories conducted the analyses using local tumor *BRCA1/2* NGS testing approaches (Table 2) and reported to reference laboratories the list of identified *BRCA1* and *BRCA2* variants interpreted as variants of unknown significance (VUS), likely pathogenic (LP), or pathogenic (P). In order to evaluate the intra-lab RTT, RL1 and RL2 agreed upon a list of 17 genotype results (from the 9 supplied samples) along with their classification as VUS (*n* = 3), LP (*n* = 1) or P (*n* = 10). Four *BRCA1/2* variants were present in tumor-derived samples (DNA1-3 and FFPE 1-3), ten in the reference samples (CC1-3), and three samples were wild-type (Table 3).

## 3. Results

### 3.1. Tissue BRCA Testing in the Bilateral RL Comparison

The rate of sequencing success for the 82 selected samples was 100% for RL1 and 98.8% for RL2 (Figure 1).

#### 3.1.1. Pathogenic and Likely Pathogenic Findings (P and LP)

Of the complete series of 100 analyzed samples, RL1 identified twenty-three (23%) P/LP BRCA mutations, 15 (65.2%) germline, and 7 (30.4%) somatic. Germline information was missing for one of the cases (PID#72) (Table 4).

Focusing on the 82 samples tested in both RLs, RL1 reported 18 (21.9%) deleterious mutations and RL2 17 (21%) (Table 4). After agreeing on the results, a total of 19 P/LP mutations reached a consensus and three major discrepancies were detected between RL1 and RL2:PID#81: *BRCA1* c.3352C>T p.(Gln1118Ter), variant present at VAF < 5%, hence, below limit of reporting of RL2 and not called. Would not be a discrepancy, since RL1 and RL2 reported correctly according to their own specifications.PID#93: *BRCA1* c.3770_3771del p.(Glu1257fs), detection failure in RL2 as a consequence of sample handling and insufficient tumor material.PID#15: *BRCA2* c.8802_8828del p.(Met2935_Gln2943del), the different criterion in the interpretation, initially classified as VUS by the RL1.

Hence, the global *BRCA* mutation frequency in our cohort was 23.17%: 12 (63.1%) germline, 6 (31.6%) somatic, and 1 (5.3%) with no germline data available. No extra deleterious mutations in *BRCA1/2* were detected in patients with germline mutations. Fifteen out of the nineteen mutations (79%) occurred in *BRCA1* and four in *BRCA2*. Two of the *BRCA1* mutations were detected in two different cases; hence, the number of different mutations was 17. Thirteen were frameshift alterations, including the four that occurred in *BRCA2*, three were nonsense (one recurrent), two were splicing alterations (recurrent) and one was missense (Table 4).

#### 3.1.2. Variants of Uncertain Significance (VUS)

A total of 11 VUS were detected by RL1 in the 82 double-tested samples. RL2 described 8 VUS, seven of which were consistent with RL1. Discordant results had diverse underlying reasons:PID#15: *BRCA2* c.8802_8828del p.(Met2935_Gln2943del), an in-frame deletion initially considered as VUS by RL1 and after consensus was reported as LP (already mentioned in the previous section).PID#13: *BRCA2* c.2771A>T p.(Asn924Ile), an interpretation disagreement, classified as VUS and likely benign by RL1 and RL2, respectively.PID#57 and PID#79: *BRCA1* c.80+6T>A and c.4986+9A>C, both variants located in intronic regions called by RL1 that was removed by the RL2 intronic threshold setting (+/− 3) in the bioinformatic pipeline.PID#73: *BRCA2* c.353G>A p.(Arg118His), a missense change not reported by RL1 due to a bioinformatic error.

In summary, after consensus, VUS were detected in 11 out of 82 patients (13.4%), 7 in *BRCA2* and 4 in *BRCA1.* Seven (64%) of the VUS alterations were missense; three were in intronic regions (*BRCA1*) and the remaining one corresponded to an in-frame deletion. Three of the patients with tumors presenting VUS carried a pathogenic mutation in *BRCA1.*

### 3.2. BRCA Ring Test Trial

Five laboratories participated in the RTT. Each of them employed different methodologies and data analysis pipelines to screen for *BRCA1/2* alterations (Table 2 and Table 5). The participating labs had different minimum variant allele frequencies (VAF) and coverage specifications for testing (Table 5).

All of the results obtained in the different labs are summarized in Table 6. A total of 14 variants were evaluated. Three samples were wild-type for *BRCA1/2* and their correct non-mutant genotype call was also included as a variant (see also Table 3). Individual sequencing analysis and clinical interpretation, as well as inter-laboratory detection and interpretation concordance for each variant, are included.

The validated 17 genotypes were sent back to the participating laboratories to allow them to re-evaluate their results and determine the reason for miscalling. Variants 1–6 were present in the six tumor samples, whereas variants 7–17 corresponded to the three reference controls.

**Variants present in tumor samples.** No individual lab obtained a 100% variant calling success when reporting the results for variants 1–6. All the participants correctly identified the wild-type samples (variants 3 and 5) and variants 1 and 8, nonsense mutations in *BRCA1*. Fails in detection in tumor samples (samples DNA_1-3; FFPE_1-3) corresponded to two frameshift mutations in *BRCA2* (variants 2 and 6) and a splicing alteration in *BRCA1* (variant 4).

Variant 2: *BRCA2* c.8802_8828del; p.(Met2935_Gln2943del), frameshift mutation not called in Lab2 as a consequence of bioinformatic filters.Variant 4: *BRCA1* c.80+6T>A, intronic VUS reported only by two laboratories; in the remaining laboratories it was missed due to the data analysis methods and filters applied for intronic sequences.Variant 6: *BRCA2* c.5351dupA; p.(Asn1784Lysfs), pathogenic mutation missed by three labs that used the Ion S5™ System (ThermoFisher Scientific) for sequencing (see Table 2). The alteration is an insertion located within a homopolymeric (polyA) region.

**Variants present in synthetic human FFPE reference samples.** The success in sample processing and sequencing was 100% for four out of the five participating labs. Lab4 reported low DNA yields from CC_1-3 local extraction. Due to this issue, CC_1 and CC_3 were processed using low DNA input amounts (below those recommended by the vendor) and no results were generated for CC_2.

Both CC_1 and CC_2 contained a total of five variants each, while CC_3 was found to be wild-type. The expected VAF for the synthetic variants presented a broad range of frequencies, some of them below the lower threshold (5%) for the majority of the laboratories (see Table 3 and Table 5). Regarding false positive results, an extra pathogenic nonsense mutation was reported in *BRCA2* with a VAF of 5.6% by one of the labs (Lab2).

With regard to discrepancies in variant classification, a total of four discrepancies affecting three variants were identified (Table 6; orange). Importantly, three of them had clinical implications (Table 6; asterisk): *BRCA1* c.3334G>T p.(Glu1112*), *BRCA2* c.8802_8828del p.(Met2935_Gln2943del) and *BRCA2* c.5351del p.(Asn1784fs) were classified as VUS instead of P/LP.

Overall, the intra-laboratory median concordance detection rate was 64.7% (35.3–70.6%) and 87.5% (75–100%) for variant classification.

Finally, reports of *BRCA* mutational tests by clinicians were also evaluated. The recommendations of Capoluongo et al. and Palacios et al. are taken as a reference [10,16]. The five report templates correctly included all data concerning patient identification and sample information (suitability of the tumor sample, neoplastic content, and statement of macro/micro-dissection). On the contrary, the methodology description sections presented high levels of heterogeneity, and relevant disclosures were missing in some cases, such as limitations in the detection of deep intronic variants or large indels, the sequencing of homopolymer regions, the description of areas of low coverage that may lead to false negative calls or the list of databases used for annotation. Concerning the interpretation of these findings and subsequent recommendations, in cases of pathogenic mutations emphasis should be placed on the need to recommend a germline study and the referral of the patient to genetic counseling.

## 4. Discussion

The approval of PARPi (olaparib, niraparib, and rucaparib) for patients with platinum-sensitive relapsed ovarian cancer (OC) has imposed the need for *BRCA* testing for proper patient management. The prevalence of g*BRCA* mutations is 18–25% in patients with high-grade OC, and an additional 8% presents s*BRCA* mutations [17,18,19,20]. In fact, both germline and somatic variants are included as predictive biomarkers of response in olaparib labeling in the EU [2,21]. Hence, the standard of *gBRCA* testing as the initial screening method has been challenged in favor of prioritizing tumor tissue DNA analysis. Tumor testing detects both germline and somatic mutations and is advantageous in terms of both time and cost, but in the absence of a paired normal sample, the variant origin may not be assigned. Therefore, any given patient with a deleterious mutation detected in a tumor sample should be referred to a genetic counseling unit so that a germline analysis of the detected variant based on observed VAF value can be performed. Several studies have evaluated the reliability of *BRCA* tumor testing compared with the germline in OC and concluded that this approach is efficient and feasible. The overall mutation rate in our study was within the range reported in other studies [6,22,23,24], as was the frequency of detected germline (63.1%) and somatic (31.6%) mutations [54–74 germline; 27–46 somatic] [5,6,23,25,26,27]. Concordance with the available *gBRCA* status data was 100%. Of note, no germline copy number variations (CNVs) were present in this particular cohort.

*BRCA* tumor testing presents relevant limitations affecting both pre-analytical (fixation process, storage conditions, age of the tumor block, tumor cellularity, tumor size, FFPE DNA extraction, and quantification, etc.) and analytical variables (library preparation, bioinformatic pipeline, variant interpretation, etc.).

This work was designed to evaluate the performance of routine real-life *BRCA* tumor testing in two settings: a two-sided cross-validation tumor *BRCA* mutation analysis of a series of 82 OC patients between two clinical reference laboratories and a ring trial involving five labs testing nine samples (six from EOC patients and three commercial controls). All labs used their own NGS methodology, data analysis, and interpretation pipelines. Pre-analytical conditions were highly homogeneous, since tumor paraffined slices were consecutively cut from a unique FFPE block per patient.

Discordance in the reported results for both sub-studies was variable. In the two-sided sub-study, discordant test results were discussed and a final genotype was agreed upon. This led to disagreement for only 3.4% of the total results, due to sample handling (one case), bioinformatic pipelines (two cases), and different variant classification criteria (two cases). In the ring trial, a perfect agreement between all participant labs was accomplished for only four genotypes (23.5%). At a more granular level, WT genotypes were correctly called by the five labs (hence, a 100% correct genotyping outcome), and point mutations and indels were correctly identified in 49% and 31% of instances, respectively.

The same NGS platform for sequencing was employed by both reference labs in the two-sided part of the study, possibly increasing the overall agreement of the obtained results. It is worth noting that *BRCA* genes include a significant amount of homopolymeric stretches, and some NGS approaches have limitations in analyzing these regions. Most of the non-reported variants in our study corresponded to indels located within low-complexity regions and this difficulty seemed to more frequently affect the labs that used an Ion S5™ System platform for sequencing (Thermo Fisher Scientific) [28,29].

Variants located within intronic regions were also a reason for non-concordance among different bioinformatic pipelines, since the distance from the canonical splicing sites cutoffs was not standardized between the labs.

The *BRCA* analysis of commercial reference samples highlighted the issues related to the limit of detection (LoD) and of reporting VAF thresholds (5–10%); for instance, CC_2 carried three variants with a VAF below 5% (*BRCA1*:c.4327C>T p.(Arg1443Ter); *BRCA2*:c.5073delA p.(Lys1691AsnfsTer15); and *BRCA2*:c.8021dup p.(Ile2675fs)). The value of LoD is particularly relevant for avoiding both false positive and false negative results. The tumor percentage of the starting sample is critical in order to maximize the detection of variants present at a frequency close to the LoD. An important limitation is a percentage of tumor cells in the sample [10,24]. In cases of low tumor cell content, pathologists should mark tumoral areas on hematoxylin-and-eosin-stained tumor slides to guide macro/micro-dissection.

Variant annotation and classification are also sources of differences among labs. A total of four clinically relevant discrepancies were detected, which could lead the oncologist to make different therapeutic decisions for the patient. This fact highlights the importance of standardizing not only the classification criteria, but also the use of specific databases and in silico prediction tools similar to those reported by the ENIGMA consortium (https://enigmaconsortium.org/ (accessed on 20 September 2022) for germline data [30]. Variants should be described as recommended by the Human Genome Variation Society (https://www.hgvs.org/ (accessed on 20 September 2022) [15] and classified according to the Standards and Guidelines for the Interpretation and Reporting of Sequence Variants in Cancer into four tiers: tier I, variants with strong clinical significance; tier II, variants with potential clinical significance; tier III, variants with unknown clinical significance; and tier IV, variants that are benign or likely benign [31]. Regarding interpretation, discrepancies affecting non-reported variants in databases remain a challenge with clinically relevant implications.

An additional limitation associated with using tumor *BRCA* analysis as a universal screening method [22,32,33,34] is the possibility of missing deleterious large rearrangements that cause CNV changes at the sub-gene scale. Chandrasekaran et al. performed parallel germline and tumor testing in OC patients, and found that all germline P variants corresponding to CNVs (with a prevalence in their series of 5/303 (1.65%)) were missed in tumor analysis [22]. Vos et al. argued that exon deletions or duplications in *BRCA* genes are a minority of the deleterious variants in OC [35] and supported tumor *BRCA* testing as a prescreening for genetic predisposition if it was performed in accredited laboratories and using validated assays. In our study, labs did not include copy number assessment as part of their testing, confirming that these alterations remain under the radar in routine tumor testing approaches.

As mentioned above, our study has limitations, one of the main ones being the lack of analysis for major rearrangements. Additionally, the genomic regions examined in this study were limited to the coding exons and flanking intronic regions. Additionally, we also highlight the threshold within the detection sensitivity for variant calling.

In summary, our study portrayed a real-life routine testing setting in hospitals. We identified the stages during analytical processing that contributed the most to the relatively low agreement among labs: bioinformatic pipelines and their pre-established settings (minimum allele frequency, splice-site cutoff intronic position, false-positive call removal, etc.) as well as differences in the criteria for the classification of variants. In conclusion, the adoption of *BRCA1/2* tumor testing will reduce the time and cost required to identify OC patients who could benefit from PARPi therapy, but critical aspects affecting the reported results are yet to be fully understood by the community, so that they may be managed to improve overall outcomes.

## Figures and Tables

**Figure 1 jpm-12-01842-f001:**
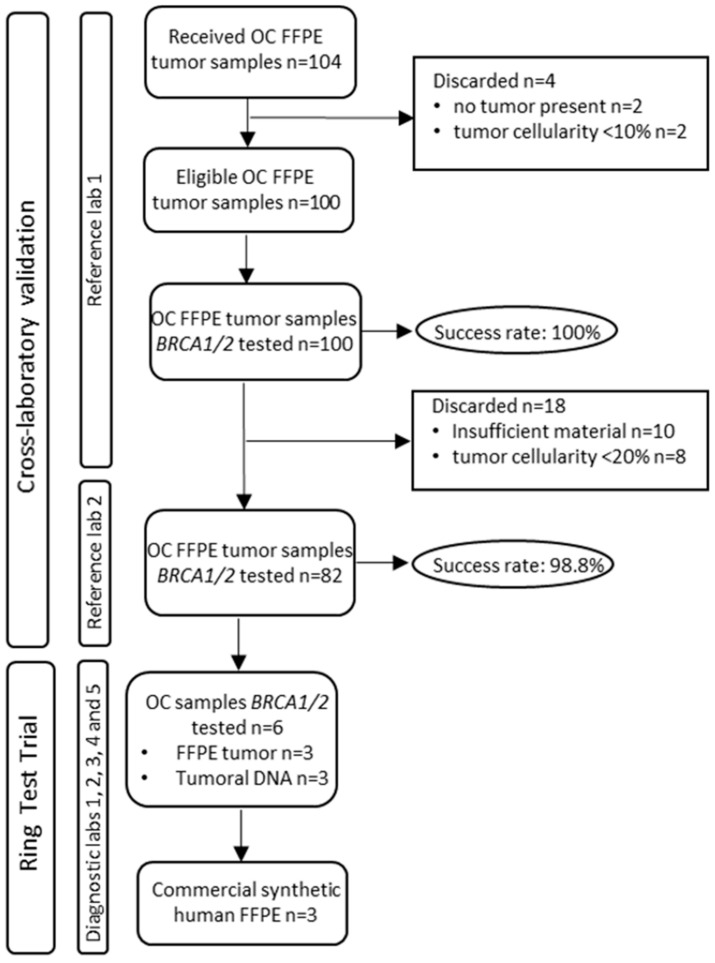
Sample selection workflow.

**Table 1 jpm-12-01842-t001:** Patient demographic and clinical characteristics.

Clinicopathological Parameters	RL1	RL2
	*n* = 100	*n* = 82
Mean age at diagnosis (range)	56 (25–84) years	56 (25–84) years
Histology		
High-grade serous	83 (83%)	69 (84.1%)
Low-grade serous	4 (4%)	2 (2.4%)
Endometrioid G1	3 (3%)	3 (3.7%)
Endometrioid G2	4 (4%)	3 (3.7%)
Endometrioid G3	0	0
Clear cells	4 (4%)	3 (3.7%)
Seromucinous	1 (1%)	1 (1.2%)
Carcinosarcoma	1 (1%)	1 (1.2%)
FIGO Stage		
I	17 (17%)	15 (18.3%)
II	9 (9%)	6 (7.3%)
III	39 (39%)	32 (39%)
IV	21 (21%)	17 (20.7%)
NA	14 (14%)	12 (14.6%)
*gBRCA*		
wt-*BRCA*	70 (70%)	58 (70.7%)
VUS-*BRCA*	13 (13%)	10 (12.2%)
mt-*BRCA*	16 (16%)	13 (15.9%)
NA	1 (1%)	1 (1.2%)

NA: not available; *gBRCA*: germline *BRCA status*; wt: wild type; VUS: variant of unknown significance; mt: mutated; RL: reference laboratory.

**Table 2 jpm-12-01842-t002:** RTT tumor *BRCA1/2* analysis strategies.

Laboratory	Chemistry	NGS-Panel	NGS-Instrument
Lab1	Multiplex-PCR	Oncomine Comprehensive Assay v3 (Thermo Fisher)	Ion S5™ System (Thermo Fisher Scientific)
Lab2	Multiplex-PCR	BRCA MASTR Plus Dx (Multiplicom)	MiSeq (Illumina)
Lab3	Hybrid Capture	Sure Select XT (Agilent)	Ion S5™ System (ThermoFisher Scientific)
Lab4	Hybrid Capture	MiniHRS (Sophia Genetics)	MiSeq (Illumina)
Lab5	Multiplex-PCR	Oncomine BRCA Research Assay (Thermo Fisher)	Ion S5™ System (Thermo Fisher Scientific)

**Table 3 jpm-12-01842-t003:** Summary of *BRCA1/2* variants present in the samples used in the RTT.

Variant Index	Sample	Variant	Clinical Classification	Expected Allelic Frequency (%)
1	DNA_1	*BRCA1:*c.3334G>T p.(Glu1112Ter)	P	49.1
2	DNA_2	*BRCA2:*c.8802_8828del p.(Met2935_Gln2943del)	LP	8.2
3	DNA_3	No pathogenic variant		
4	FFPE_1	*BRCA1:*c.80+6T>A	VUS	47.4
5	FFPE_2	No pathogenic variant		
6	FFPE_3	*BRCA2:*c.5351dupA p.(Asn1784Lysfs)	P	20.1
7	CC_1	*BRCA2:*c.5351del p.(Asn1784fs)	P	40
8	CC_1	*BRCA1:*c.4327C>T p.(Arg1443Ter)	P	32.5
9	CC_1	*BRCA2:*c.5073del p.(Lys1691fs)	P	32.5
10	CC_1	*BRCA2:*c.8021dup p.(Ile2675fs)	P	10
11	CC_1	*BRCA1:*c.1303G>T p.(Asp435Tyr)	VUS	7.5
12	CC_2	*BRCA2:*c.5351del p.(Asn1784fs)	P	10.2
13	CC_2	*BRCA1:*c.4327C>T p.(Arg1443Ter)	P	**3.9**
14	CC_2	*BRCA2:*c.5073delA p.(Lys1691AsnfsTer15)	P	**3.1**
15	CC_2	*BRCA2:*c.8021dup p.(Ile2675fs)	P	**4.5**
16	CC_2	*BRCA1*:c.1303G>T p.(Asp435Tyr)	VUS	8.2
17	CC_3	No pathogenic variant		

P: pathogenic; LP: likely pathogenic; VUS: variant of unknown significance. Expected allelic frequencies below 5% are in bold.

**Table 4 jpm-12-01842-t004:** Pathogenic and likely pathogenic BRCA1/2 variants detected in EOC tumors.

Patient ID	Description	RL1			RL2			Germline
		VAF (%)	Reads	Class	VAF (%)	Reads	Class	
7	NM_007294.3(*BRCA1*):c.3627dupA p.(Glu1210Argfs) rs80357729	71	4287	P	46	361	P	Yes
11	NM_007294.3(*BRCA1*):c.1674del p.(Gly559fs) rs80357600	59	13,641	P	52	362	P	Yes
13	NM_007294(*BRCA1*): c.2843_2849del p.(Gly948Valfs*50)	26	5823	P	13	313	P	No
14	NM_007294.3(*BRCA1*):c.66_67AG p.(Glu23fs) rs80357914	70	13,914	P	64	342	P	Yes
15	NM_000059.3(*BRCA2*):c.8802_8828del p.(Met2935_Gln2943del)	8.2	6380	VUS	6	870	P	No
19	NM_007294.3(*BRCA1*):c.115T>A p.(Cys39Ser) rs80357164	75	3148	P	78	125	P	Yes
21	NM_007294.3(*BRCA1*):c.3331_3334del p.(Gln1111fs) rs80357701	68	557	P	57	65	P	Yes
28	NM_007294.3(*BRCA1*):c.3648dupA p.(Ser1217Ilefs) rs80357902	80	5391	P	72	556	P	No
31	NM_007294.3(*BRCA1*):c.3752_3755GTCT p.(Ser1253fs) rs80357868	55	4391	P	40	292	P	Yes
35	NM_000059.3(*BRCA2*):c.715dup p.(Ser239fs) rs431825350	51	5313	P	52	257	P	Yes
48	NM_007294.3(*BRCA1*): c.3334G>T p.(Glu1112Ter)	49	18,219	P	53	704	P	No
62	NM_000059.3(*BRCA2*):c.1128del p.(Phe376fs) rs80359263	73	4935	P	57	393	P	Yes
70	NM_000059.3(*BRCA2*):c.5351dupA p.(Asn1784Lysfs) rs80359507	21	16,034	P	23	220	P	No
71	NM_007294.3(*BRCA1*):c.845C>A p.(Ser282Ter) rs786203027	42	298	P	52	639	P	Yes
72	NM_007294.3(*BRCA1*): c.5578del p.(His1860Thrfs*?)	48	2399	P	36	110	LP	NA
79	NM_007294.4(*BRCA1*):c.211A>G p.(Arg71Gly) rs80357382	71	6629	P	72	185	P	Yes
80	NM_007294.4(*BRCA1*):c.211A>G p.(Arg71Gly) rs80357382	71	283	P	69	131	P	Yes
81	NM_007294.3(*BRCA1*):c.3352C>T p.(Gln1118Ter) rs397507215	15	3636	P	UR			No
93	NM_007294.3(*BRCA1*):c.3770_3771del p.(Glu1257fs)	60	3503	P	UR			Yes
5	NM_000059.3(*BRCA2*):c.9026_9030del p.(Tyr3009fs)	45	994		NT			Yes
27	NM_007294.3(*BRCA1*):c.3627dupA p.(Glu1210Argfs)	45	3351		NT			Yes
36	NM_000059.3(*BRCA2*):c.6275_6276del p.(Leu2092fs)	53	21,450		NT			Yes
63	NM_007294.3(*BRCA1*):c.1A>G p.(Met1Val)	18	701		NT			No
110	NM_000059.3(*BRCA2*): c.3022del p.(Ser1008Alafs*35)	94	4321		NT			No

P: pathogenic; LP: likely pathogenic; VUS: variant of unknown significance; VAF: variant allelic frequency; UR: unreported; NT: non-tested; RL: reference laboratory; *: TERMINATION (Ter).

**Table 5 jpm-12-01842-t005:** RTT NGS bioinformatic pipelines.

RTT Laboratory	Data Analysis Tools/Pipelines	VAF	Minimum Coverage	Intron Flanking Region	Variant Annotation Databases
Lab1	Ion ReporterTM Software Version 5.10	5%	500×	±10 bp	ClinVar, Varsome; COSMIC
Lab2	MASTR Reporter 1.3.0	5%	1000×	No	ClinVar; *BRCA* Exchange
Lab3	novocraft V3.07.01, bamtools-2.4.1, VCFtools (0.1.15), bedtools v2.26.0-40, samtools 1.8, picardtools 2.8.3, ensembl vep release 94, CONTRA.v2.0.8, gatk-3.4.46	10%	20×		NCBI, ClinVar, Ensembl, *BRCA* Exchange, cBioPortal
Lab4	Sophia DDM v3-Sophia Genetics	5%	500×		ClinVar, COSMIC, dbSNP, EXAC, g1000, ESP, EpiCov, GnomAD,
Lab5	Ion Reporter Software Version 5.16	5%	100×	±100	dbSNP, BIC database, *BRCA* Exchange, *BRCA* Mutation Database

RTT: Ring Test Trial; VAF: variant allelic frequency.

**Table 6 jpm-12-01842-t006:** Summary of *BRCA1/2* variants and results obtained in the RTT.

Variant	Type of Variant	Sample	Lab1	Lab2	Lab3	Lab4	Lab5	Detection Concordance to Reference Genotype (%)	Interpretation Concordance to Reference Genotype (%)
1	nonsense	DNA_1				*		100%	60%
2	frameshift	DNA_2				*		80%	75%
3	wt	DNA_3						100%	100%
4	splicing	FFPE_1						40%	100%
5	wt	FFPE_2						100%	100%
6	frameshift	FFPE_3						40%	100%
7	frameshift	CC_1						20%	100%
8	nonsense	CC_1						100%	100%
9	frameshift	CC_1						60%	100%
10	frameshift	CC_1						20%	100%
11	missense	CC_1						20%	100%
12	frameshift	CC_2					*	50%	50%
13	nonsense	CC_2						50%	100%
14	frameshift	CC_2						25%	100%
15	frameshift	CC_2						0%	100%
16	missense	CC_2						25%	100%
17	wt	CC_3						100%	100%

Green: concordance in detection and interpretation; red: no reporting; orange: concordance in detection but not in interpretation; grey: no results delivered by the lab; *: discrepancy with clinical relevance.

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
