# Peer review of "Laboratory Cross-Comparison and Ring Test Trial for Tumor BRCA Testing in a Multicenter Epithelial Ovarian Cancer Series: The BORNEO GEICO 60-0 Study"

_jpm, 2022, doi:10.3390/jpm12111842_

Round 1
Reviewer 1 Report
This study mainly focuses on the identification and interpretation of BRCA sequencing data, which is of great significance for the research of oncology. However, some content needs to be adjusted before the article can be accepted.
Major points:
1. In Materials and Methods, the NGS description needs to be described in detail, including sequencing quality control standards, the number of reads supported by sequencing sites, etc.
2. In Table 1, the statistical comparison of RL1 and RL2 groups should be included, please add it.
3. Please add the limitations of this study, such as the lack of LOH analysis, etc.
Minor points:
1. Some abbreviations in the article need to be reconfirmed, the first time should not directly use abbreviations, eg. "NGS" in Line 58.
Author Response
This study mainly focuses on the identification and interpretation of BRCA sequencing data, which is of great significance for the research of oncology. However, some content needs to be adjusted before the article can be accepted.
Major points:
- In Materials and Methods, the NGS description needs to be described in detail, including sequencing quality control standards, the number of reads supported by sequencing sites, etc.
We have now added a more detailed description about NGS in the Material and Methods section, especially for RL1 and RL2. Concerning Ring Test Trial participating laboratories, a summary with the main features about their NGS strategies and pipelines can be found in Table 2 and Table 5. The methodology for each group could be described in detail if required by the reviewer.
- In Table 1, the statistical comparison of RL1 and RL2 groups should be included, please add it.
We appreciate the reviewer’s comment, however, Table 1 is just a descriptive table including the demographic and clinical characteristics of the patients included in the study with no information on tumor BRCA mutations. The BRCA mutations collected in table 1 refers to the germline status, whose information has been provided by the centres involved in the study. Data about tumour mutations detected by RL1 and RL2 are detailed in Table 4 included in the Results section.
- Please add the limitations of this study, such as the lack of LOH analysis, etc.
Certainly, there were limitations to this study. This point has been addressed in the Material and Methods description (“The analysis did not include large rearrangements. The sensitivity limit was set at 5% for point variants and 10% for insertion or deletion variants.”) as well and in the Discussion section as follows “In our study, labs did not include copy number assessment as part of their test, confirming these alterations remain under the radar in routine tumor testing approaches”. However, a specific paragraph referring to the limits of our study has been added.
Extensive editing of English language and style required
MDPI language editing service has performed the English language and style review.
Minor points:
- Some abbreviations in the article need to be reconfirmed, the first time should not directly use abbreviations, eg. "NGS" in Line 58.
We apologize for the mistake and we have included the complete definition for the abbreviation in line 58 (Abstract). We have reviewed the rest of the text and we hope we have not left any other undefined abbreviations.
Reviewer 2 Report
In this paper, the authors investigated the concordance of BRCA testing of ovarian cancer samples across different laboratories to determine inter-laboratory reproducibility of somatic BRCA testing results. Due to the different sequencing equipments, techniques and variant calling threshold, the authors did recover the existence of discrepancy in the somatic BRCA results.
This research is of great clinical importance for standardizing the NGS analytical approaches used to call somatic BRCA variance. It raises awareness of the potential bias of the results due to testing batch effect.
I would be curious to know if this type of certain discrepancy was previously observed in other BRCA testing, for example, for breast cancer. In this case, the authors could comment on the background of the somatic BRCA testing in a bit more detailed format.
In Table3, the Expected Allelic Frequency (%) column name already indicated that the value will be in percentage. But each individual value is still labeled with percent sign, which is repetitive.
In Table 4, this is an empty read count in RL2 for ID=13.
Author Response
In this paper, the authors investigated the concordance of BRCA testing of ovarian cancer samples across different laboratories to determine inter-laboratory reproducibility of somatic BRCA testing results. Due to the different sequencing equipments, techniques and variant calling threshold, the authors did recover the existence of discrepancy in the somatic BRCA results.
This research is of great clinical importance for standardizing the NGS analytical approaches used to call somatic BRCA variance. It raises awareness of the potential bias of the results due to testing batch effect.
We really appreciate the comments of Reviewer 2.
I would be curious to know if this type of certain discrepancy was previously observed in other BRCA testing, for example, for breast cancer. In this case, the authors could comment on the background of the somatic BRCA testing in a bit more detailed format.
According to this comment, we have not identified any similar study to ours in other tumour types referring to BRCA testing. However, aside from the differences due to the initial sample, we believe that many of our findings can be extended to the analysis of these mutations in other tumour types.
In Table3, the Expected Allelic Frequency (%) column name already indicated that the value will be in percentage. But each individual value is still labeled with percent sign, which is repetitive.
We have reviewed the Table 1 and eliminated the percent sign.
In Table 4, this is an empty read count in RL2 for ID=13.
We have reviewed the NGS data from RL2 and included the missing value for ID=13 read count.
Round 2
Reviewer 1 Report
The newly revised manuscript meets the requirements for publication.